# Transmembrane Homodimers Interface Identification: Predicting Interface Residues in Alpha-Helical Transmembrane Protein Homodimers Using Sequential and Structural Features

**DOI:** 10.3390/ijms26094270

**Published:** 2025-04-30

**Authors:** Bander Almalki, Li Liao

**Affiliations:** Department of Computer and Information Sciences, University of Delaware, Smith Hall, 18 Amstel Avenue, Newark, DE 19716, USA; alathwny@udel.edu

**Keywords:** transmembrane homodimers, dimerization, interface resides prediction, machine learning, molecular dynamics

## Abstract

Most bitopic transmembrane proteins associate with one another through interface residues to form dimers, which facilitate or activate specific cellular functions. Therefore, accurately identifying interface residues in a given dimer is crucial for understanding its function and has been a challenging pursuit for many computational methods. These methods can be broadly categorized into two approaches: general-purpose ones for dimerization and specialized ones for interface residues. In this study, we develop a machine learning method that integrates both approaches by integrating sequential and structural features extracted from predicted structures and various domains. The results from cross-validation on a benchmark dataset show that our method, despite utilizing significantly fewer features, outperforms the state-of-the-art methods by more than three percentage points in performance, as measured by the F1 score. Furthermore, we evaluated the performance of the proposed model on a benchmark dataset as compared to the state-of-the-art multimeric structure predictors, including RoseTTAFold2, AlphaFold2Multimer, and PREDDIMER. The results show the superiority of the proposed model by outperforming all the other models, highlighting the effectiveness of integrating both structural and sequential features within the proposed framework.

## 1. Introduction

Proteins are large molecules that play crucial roles in various living organisms, from providing cellular structure, such as actin, to protecting the body from foreign particles like viruses and bacteria, as seen in immunoglobulin G [1]. Transmembrane (TM) proteins, integral components of the cell membrane, represent a significant portion of all known proteins. For example, it is estimated that nearly 20–30% of all proteins encoded by most organisms’ genomes are TM proteins [2]. These proteins can function as receptors, such as G-protein-coupled receptors [3] and receptor tyrosine kinases (RTKs) [4], or as transporters like glucose transporter (GLUT1) [5] and sodium–potassium pump (Na^+^/K^+^ ATPase) [6]. A fundamental characteristic of monomer proteins is their ability to associate into complexes, which is critical for structural stabilization and functional activation [7]. In TM proteins, two single-pass (bitopic) TM monomers can associate to form a dimer through a process called dimerization, which is essential for stabilizing their structures and activating downstream cellular functions [8]. For instance, the association of two bitopic RTKs, triggered by ligand binding, can induce a conformational change in the protein, activating a downstream signaling pathway within the cell [9]. Incorrect dimerization can have detrimental impacts on the cell by impairing signaling pathways, potentially leading to oncological diseases [10]. Therefore, the accurate determination of homodimer structures is crucial for understanding their functions and developing targeted therapies.

The significance of protein structure determination stems from the fundamental principle that structure dictates function [11]. Despite the rapid increase in the number of resolved protein structures, a considerable gap persists between the number of known protein sequences and their corresponding experimentally determined structures. For instance, UniProt contains over 245 million sequences, while only approximately 222,000 structures are deposited in the Protein Data Bank (PDB). TM proteins are no exception and are still under-represented in the PDB. For example, according to the latest version of the PDBTM database [12], only 10,932 of the known structures are TM proteins. This scarcity of known TM protein structures in the PDB can be attributed to two primary factors. First, the high cost of experimental methods limits the number of structures that can be determined [13,14]. Second, the extraction and isolation of TM proteins present substantial challenges. For example, the extraction method can induce distortion to the protein, which can significantly compromise the accuracy of the resulting structure. This issue is particularly relevant for methods such as X-ray crystallography, which remains the predominant technique for deposited structures in the PDB. In X-ray crystallography, the TM protein must be isolated from the cell membrane to be scanned, a process that can lead to protein deformation and ultimately produce an inaccurate structure [1,15]. The disparity in available structures is even more pronounced for TM protein homodimers compared to globular homodimers or even TM protein monomers. These challenges highlight the need for advanced computational approaches capable of accurately predicting TM protein homodimer structures and identifying key interface residues.

Numerous studies have been devoted to investigating the dimerization of bitopic TM proteins. These studies can be broadly categorized into two primary approaches: a general-purpose approach, which focuses on predicting the dimeric structure, and a specialized approach dedicated to identifying interface residues with greater precision. Among the first approaches, PREDDIMER [16] and TMDOCK [17] are prominent examples. PREDDIMER is a surface-based modeling framework that predicts the conformations of alpha-helical bitopic TM proteins by employing a molecular hydrophobicity potential (MHP) approach to reconstruct the dimer’s conformation, followed by molecular dynamics (MDs) relaxation to predict the final structure. TMDOCK, a free energy-based approach, uses alpha-helical TM bitopic amino acid sequences as input, applies threading to generate a 3D structure of the homodimer, and utilizes free energy minimization techniques to select the optimal conformation. Recently, large attention-based models such as AlphaFold2 [18] and RoseTTAFold [19] have demonstrated remarkable success in predicting the structure of monomeric proteins. Enhanced versions of these models, AlphaFold2-Multimer [20] and RoseTTAFold2 [21], have been introduced for multimer structure prediction and could potentially be applied to TM protein dimers. Once the dimer structure is predicted, the interface residues can be consequently identified by applying a threshold on the distance between two residues across the two monomers. However, the actual performance of these models in predicting the interface residues in TM protein homodimers has not yet been tested or verified, an issue that is specifically addressed in the present study. Unlike the general-purpose approach, the specialized approach does not aim to predict the overall dimer structure but instead focuses on more accurately identifying interface residues, which can facilitate the design of mutants for modulated dimerization. EFDOCK [22] exemplifies a model using a pure statistical approach to predict inter-chain residue contacts in TM helical proteins. The model operates on the hypothesis that the coevolutionary signal of TM protein residues that form inter- and intrachain contacts is stronger than that that forms only intrachain contacts. EFDOCK takes the amino acid sequences of TM proteins as input, extracts coevolutionary features using EVfold [23], and filters the predicted contacts using the LIPid-facing Surface (LIPS) [24] and Direct Interaction (DI) scores sequentially. To date, few models have employed machine or deep learning techniques to address this problem. One notable exception is THOIPA [25], which uses an extreme randomized tree algorithm to predict the interface residues in alpha-helical TM proteins forming homodimers. THOIPA has identified key characteristics of interface residues, demonstrating that they generally tend to be more conserved, polar, deeply embedded in the lipid bilayer, and rich in small-xxx-small motifs. However, THOIPA is primarily effective in predicting the top n interface residues, where 1≤n≤5, and has shown modest performance in predicting the entire interface region compared to PREDDIMER and TMDOCK.

In this study, we pursue two primary objectives: (1) to systematically assess the predictive accuracy of existing methods for interface residue identification in TM protein homodimers, including molecular dynamics-based models, attention-based models, large protein language models, and machine/deep learning approaches; and (2) to develop a novel machine learning model that integrates multiple predictive strategies. Specifically, our proposed model leverages sequence-based features extracted from a large protein language model, incorporates structure-based features extracted from a surface-based modeling approach, and integrates key physicochemical properties to enhance the accuracy of interface residue prediction in alpha-helical TM protein homodimers.

The results show that, despite the significant advancements achieved by recent multimer structure models such as AlphaFold2-Multimer and RoseTTAFold2—both of which demonstrate the ability to predict various monomer structures with near-atomic accuracy—they are still inferior to the helical surface-based models, such as PREDDIMER, in predicting interface residues in TM protein homodimers (Table 1). To the best of our knowledge, this is the first study to explicitly highlight these model limitations in this context. Furthermore, the results reveal that when it comes to identifying interface residues, specialized approaches, such as THOIPA and its like, outperform the general-purpose approaches, such as AlphaFold2-Multimer, RoseTTAFold2, and PREDDIMER (Table 1 and Table 2). Moreover, we demonstrate that our proposed model, TMH-ID, which uses a reduced number of features by incorporating sequential and structural features from various domains, surpasses both state-of-the-art machine learning models and large protein language models in accurately predicting interface residues in TM protein homodimers (Table 2).

## 2. Materials and Methods

### 2.1. Dataset

In this study, we adopt the dataset introduced in [25], which contains 50 TM homodimer proteins with a resolution better than 3.4 Å. The dataset is divided into three subsets based on the experimental method employed for structure determination (Crystal, NMR, and ETRA) (Figure 1). The Crystal and NMR subsets consist of 21 and 8 proteins, respectively, as deposited in the PDB database. On the other hand, the ETRA subset contains 21 homodimers identified through scanning mutagenesis in combination with the ToxR assay. In the ETRA subset, a total of 263 mutations at 203 positions were experimentally assessed to identify the interface residues of the 21 homodimers.

### 2.2. Definition of Interface Residues

Various studies employ different thresholds to distinguish between interface and non-interface residues in TM protein homodimers. However, to ensure a fair comparison, we adopt the same threshold as applied in [25]. Specifically, a residue i in helix 1 is classified as an interface residue if the minimum distance between any of its heavy atoms and any other heavy atom of a residue in helix 2 is less than 3.5 Angstrom (Å). For the Crystal and NMR subsets, this threshold is used. On the other side, the dimerization disruption cut-off value of 0.24 is used for the ETRA subset.

### 2.3. Multiple Sequence Alignment Extraction

Extracting Multiple Sequence Alignment (MSA) is an initial step toward extracting sequence-related features such as the Position Specific Scoring Matrix (PSSM) and Direct Coupling Analysis (DCA) features. However, while extracting MSAs for monomers is direct and can be a relatively easy task using searching tools such as BLAST [26] and HHblits [27], extracting MSAs for dimers presents notable challenges and might not be trivial. That is, the sequences of bitopic TM protein monomers are usually very short, which makes the extraction of high-quality MSAs hard. Moreover, in homodimers, the two monomer units possess identical sequences, which complicates the extraction of high-quality MSAs when merging the two sequences. In addition, this sequence symmetry makes it difficult to differentiate between inter-chain and intra-chain contacts. A proposed solution by [28] is that only long-range contacts are considered as inter-residue contacts to distinguish them from intra-residue contacts. However, this raises another issue: some residues participate in both inter- and intrachain contacts.

In this study, to extract the MSAs, we follow the methodology outlined in [25] where the monomer sequence, along with the concatenated 20 residues from both N and C terminals, is used to produce the MSAs of each dimer in the dataset. This approach aims to compensate for the relatively short length of TM protein monomers and to enhance the number of MSAs available for extracting sequence-based features essential for predicting interface residues in TM protein homodimers. For all the homodimers in the dataset, HHblit was used to generate the MSAs using the UniClust30 database and the following parameters. Number of iterations = 2 and E-value cutoff = 0.001. Note that after generating the MSAs of all the dimers, the concatenated 20 residues are chopped off, and only the TM domain sequence is kept.

### 2.4. Sequence-Based Features

#### 2.4.1. Extracting Coevolutionary Features

Coevolutionary features characterize how two residues in a protein co-evolve over the evolutionary phase. In intra-chain residue contact prediction, this feature has been shown to be very informative as a strong signal of contact between residues [23,29,30]. Recently, Coevolution has shown success in the inter-chain contacts too [22,31]. For example, in [25], the authors show that interface residues have a stronger indication of evolution than non-interface residues.

While various Direct Coupling Analysis (DCA) techniques have been used extensively to extract the coevolutionary features, recently large protein language models have shown their superiority in the field. In this study, MSA Transformer [32] is utilized to compute coevolutionary scores for each residue pair within TM homodimers. The model employs an unsupervised masking approach as an objective function and the maximum likelihood as a loss function, as follows:$$L_{MLM}\left(x,\theta \right)=\sum_{(m,i)\in mask}\log P\left(x_{mi}|\bar{x};\theta \right)$$
where *x* represents the MSA loss and $\bar{x}$ the masked MSA. The model then applies the tied row attention mechanism to capture coevolutionary relationships between residue pairs, as follows:$$\sum_{m=1}^{M}\frac{Q_{m}K_{m}^{t}}{\lambda (M,d)}$$
where *M* is the number of rows, *d* is the hidden dimension, and Q,k are the query and key matrices, respectively. The square-root normalization $\lambda \left(M,d\right)=\sqrt{Md}$ is used by the MSA Transformer to calculate the tied row attention. While the model can take various numbers of MSAs as input, only the top 128 sequences are used in this study due to resource limitations. The final output of the model is an L∗L scoring matrix that quantifies the correlation between each residue pair. In this study, for residue i in helix 1 and residue j in helix 2, the following scores are used to predict the interface residue in helix 1, $\left(i,j_{1}\right),\;\left(i,j_{2}\right),\;\left(i,j_{3}\right),\ldots ,\left(i,j_{n}\right)$, where *n* is the length of the monomer. Since different proteins can have different lengths, and to standardize this feature for all proteins, Principal Component Analysis (PCA) with *n* = 10 is used. Note that the PCA led to better accuracy compared to other approaches, such as the mean, max, median, or sum of the coupling score.

#### 2.4.2. TM Monomer Sequence Motifs

Three sequence motifs are used as features in this study: GxxxG, small-xxx-small, and polar-xxx-polar motifs. The GxxxG motif, characterized by a sequence pattern within TM helices where “G” denotes glycine and “xxx” represents any three amino acids, plays a major role in the dimerization of TM proteins [33]. The small size of glycine in this motif allows for close packing of helices, thereby promoting tight interactions and efficient helix–helix association [34]. These interactions are critical for the structural stability and functional integrity of various membrane proteins, including receptors and channels [35]. This motif has been shown to be over-represented in TM dimers and is one of the most informative features implying dimerization [25,33,34,35]. In addition to the GxxxG motif, small-xxx-small and polar-xxx-polar scores are also incorporated. In these motifs, “small” refers to amino acids with small side chains, while “polar” denotes amino acids with known polar characteristics. These latter motifs have also been demonstrated to be rich in the interface residues of TM homodimers [25]. Each monomer’s residue is assigned a binary value of 1 or 0, indicating the presence or absence of the respective motif.

#### 2.4.3. LIPS Scores

The LIPid-facing Surface (LIPS) is a computational tool designed for predicting the helix-lipid interface in TM helices [24]. In its original implementation, the tool was evaluated on a dataset of 18 TM proteins with numbers of helices ranging from 4 to 24. LIPS takes MSAs as input, divides the helix into seven helical faces, and outputs the average lipophilicity, entropy, and lipophilicity ∗ entropy scores of each surface. A high LIPS score suggests increased lipid exposure of the corresponding helical face. These three scores are incorporated as features in our pipeline.

#### 2.4.4. Structure-Based Features

Structure-based features have been shown to be very useful in intra-molecular contacts [36], so in this study we explore how structure-based features may help with inter-molecular contact. Specifically, we extract the homodimer structure by running the alpha-helical-surface-based model PREDDIMER [37]. PREDDIMER takes the monomer’s sequence of the TM alpha-helical protein as input and produces the most probable 3D dimeric structure as output. This process involves four key steps. First, the reconstruction of the dimer conformations with the help of PLANTUM software (v1.2025.2) [38], which evaluates the hydrophobicity properties of the surface of each helix. Based on this, the rotational angle of each helix and the number of crossing angles can be calculated. Second, since many confirmations might result from the first step, the packing quality of the helices is evaluated using a scoring function (FSCOR). Third, similar resulting structures are clustered together based on Root-Mean-Square-Deviation (RMSD) and inter-helical contact thresholds. After that, the output structures are ranked using the FSCOR. Finally, the output structure can be further optimized using MD simulation of the dimer in the membrane.

In this study, we incorporate the minimum distance of heavy atoms between each residue *i* in helix 1 and all the other residues in helix 2 (Figure 2) as a new feature. This structural feature is incorporated alongside other sequence-based features to enhance the prediction of interface residues.

## 3. Results

### 3.1. Comparing TM Proteins Surface-Based vs. Attention-Based Models

With the unprecedented success of current sophisticated attention-based deep learning models in monomer structure prediction, such as AlphaFold2 [18] and RoseTTAFold2 [19], new versions have been proposed for predicting multimer structures, including dimers. However, the assessment of these models’ accuracy on TM protein homodimers is still limited.

**Figure 3 ijms-26-04270-f003:**
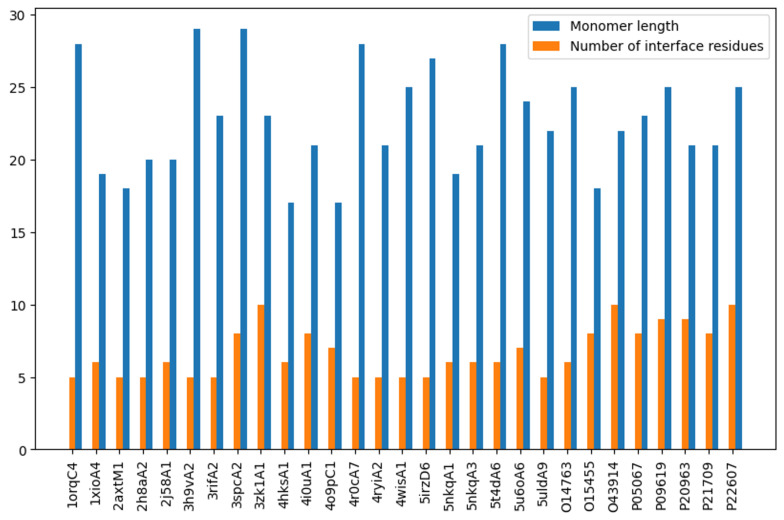
The length of the sequence (blue) compared to the number of interface residues (orange) (threshold = 3.5) in Crystal and NMR subsets. The bar chart shows the skewness of the data, with number of interface residues being less than 10 in each dimer.

In this study, we evaluate how well the state-of-the-art multimer attention-based models, RoseTTAFold2 [21] and AlphaFold2Multimer [20], can predict the correct interface residues in TM protein homodimers. Then, we compare these models with the α-helical surface-based model PREDDIMER [16].

#### 3.1.1. Extracting the Predicted Structures

The predicted structures of each homodimer in the Crystal and NMR datasets were generated using the models described above. For AlphaFold-Multimer and RoseTTAFold2, we employed their publicly available Colab implementations [39] to obtain the predicted 3D structures. Both models accept single- or multichain protein sequences as input, and output the corresponding tertiary structure. For homodimer predictions, where the two chains are identical, the sequences were separated by a colon to indicate chain duplication. In the case of homodimer structure prediction, since the two chains have the exact same sequence, they should be separated by a colon. In the case of PREDDIMER, the model takes a single-pass transmembrane (bitopic) protein sequence as input and returns a ranked list of predicted dimer structures. However, this model is limited to small sequences with a number of residues ranging from 20 to 35 amino acids. To accommodate sequences shorter than 20 residues, we appended five neighboring residues from both the N- and C-termini. These appended residues were subsequently removed from the predicted structures to ensure a consistent and fair structural comparison.

#### 3.1.2. Extracting Interface Residues

Interface residues were identified from predicted structures by first computing the pairwise distances between all heavy atoms of residue pairs across the structure. A residue was defined as an interface residue if any of its heavy atoms were within 3.5 Å of a heavy atom from a residue located on a different helix. This distance-based threshold was applied uniformly across all predicted structures to determine interfacial contacts. For ground truth comparison, interface residues were extracted from the corresponding experimentally resolved PDB structures using the same 3.5 Å threshold.

#### 3.1.3. PREDDIMER Outperforms RoseTTAFold2 and AlphaFold2Multimer

To compare PREDDIMER vs. RoseTTAFold2 and AlphaFold2Multimer, the F1 score is used. True positive indicates that the structure predicted by the model successfully identifies an interface residue in the dimer. Surprisingly, we found that the simple α-helical surface-based model PREDDIMER, which does not require training, outperforms the other two attention-based models significantly in predicting the interface residues in TM homodimers.

Among the three predictors, RoseTTAFold2 is the least predictive, with a mean F1 score of 0.137 (Table 1). For a few homodimers, the model failed to even predict the correct 2D structure. For example, in homodimer 5nkqA3 [40] (Figure 4), RoseTTAFold2 predicted random coils instead of an α-helical structure. Moreover, we noticed that for most (17 out of 30) dimers, the model failed to predict any correct interface residue (Figure 5). However, we found that for a few homodimers that have more than two chains in the PDB Database, such as 2axt and 5u6o, RoseTTAFold2 was better than the other two models. On the other side, AlphaFold2Multimer was the second-best model in predicting the interface residues in TM homodimers with a mean F1 score of 0.302 (Table 1). The model successfully predicted the 2D structure of all the dimers. Out of the 30 homodimers in the Crystal and NMR sets, AlphaFold2Multimer failed to identify any interface residues in only 4 and surpassed the other models in 9 (Figure 5).

Compared with RoseTTAFold2 and AlphaFold2Multimer, PREDDIMER achieved the best mean precision, recall, and F1 scores. The model scores 0.378 on mean F1 score, outperforming AlphaFold2Multimer by more than 7% (Table 1). Being an α-helical surface-based predictor, PREDDIMER successfully identified all the dimers’ 2D structures. Moreover, the model is the only predictor that exceeded an F1 score of 0.7 on individual dimers (Figure 5). In terms of F1 score on individual dimers, PREDDIMER surpassed RoseTTAFold2 and AlphaFold2Multimer in 22 and 18 homodimers, respectively.

### 3.2. TMH-ID Model

Inspired by the success of PREDDIMER in predicting the interface residues in TM homodimers, we developed the Transmembrane Homodimers Interface Identification (TMH-ID) model. TMH-ID (Figure 6) is a machine learning model that combines both sequence-based features and helix-interface-based features to improve the prediction accuracy of interface residues in TM homodimers. The model takes the monomer sequence as input and generates the predicted interface residues as output. The monomer sequence is used to extract various motifs, MSAs using HHblit, and structural atomic distances using PREDDIMER. The MSAs are used to extract the LIPS scores using the LIPid tool and the coevolutionary feature using the MSA Transformer. These features are then used as input to a logistic regression classifier to predict the interface residues.

#### 3.2.1. Comparing TMH-ID to ProteinBert, MSA Transformer, and THOIPA

Using all 50 homodimer proteins from the three sets—Crystal, NMR, and ETRA—we evaluate the prediction accuracy of ProteinBert, MSA Transformer, and THOIPA. Then, we compare these models with our proposed model, “TMH-ID”. Even with a small number of features, our results demonstrate the superiority of TMH-ID on the TM homodimer interface residue prediction task compared to the other models.

For a fair comparison, all the models are trained and tested using a simple logistic regression model, as implemented by the Sicket-learn package [41] with the following hyperparameters. First, to make sure that the model has enough iterations to converge, the max-iter parameter is set to 1000. Second, to mitigate the skewness in the data (Figure 3), where the number of interface residues is only 304 compared with 787 non-interface ones, the class weight is set to balanced. Finally, five-fold cross-validation is used to evaluate each model, and the mean precision, recall, and F1 scores are reported.

ProteinBert [42] is an attention-based large pretrained trainable protein language model pretrained on approximately 106 million proteins and consists of 16 million parameters. Its architecture is inspired by the well-known large language model BERT. The model takes a protein sequence as an input and outputs an embedding vector of length 1024 per residue. In this study, to compare ProteinBert with our model, the monomer’s sequence of the homodimer is used to extract the embeddings. Then, these embeddings are used to train and test a model to predict the interface residue TM.

MSA Transformer [32] was trained on approximately 26 million MSAs with 100 million trainable parameters. In this study, the final output score of the model plus the LIPS scores are used to train and test a logistic regression model to predict the interface residue. The results show that with an F1 score of 37.8, MSA Transformer is better than ProteinBert, but lags behind THOIPA and TMH-ID. However, it is worth noting that if only the MSA Transformer score is used, the F1 score drops to 35.5. A possible reason for this performance drop is that the model focuses only on extracting the coevolutionary information between the residues but ignores other critical features for homodimer association, such as hydrophobicity and lipophilicity.

THOIPA uses a total of 140 features, including Position Specific Scoring Matrix (PSSM), Direct Coupling Analysis (DCA) scores, Lipophilicity, hydrophobicity, and other handcrafted features. In this study, for comparison, the same features are used to train and test a model to predict the interface residues in a dimer. The results demonstrate that THOIPA scores the second best after our model, TMH-ID, with an F1 score of 40.9. This signifies the importance of helical surface-based features in improving prediction accuracy.

TMH-ID uses a total of 17 features per residue only, making it much faster than all the other models. The merging of sequence-based and helical surface-based features boosts the model’s accuracy. TMH-ID achieves a mean F1 score of 43.9, outperforming ProteinBert, MSA Transformer, and THOIPA by 10%, 6%, and 3%, respectively.

#### 3.2.2. Comparing TMH-ID to PREDDIMER, AlphaFold2Multimer, and RoseTTAFold2

To assess the performance of the proposed model, TMH-ID, in comparison to leading interface prediction approaches, we evaluated it against two state-of-the-art attention-based multimer structure predictors—AlphaFold2-Multimer and RoseTTAFold2—as well as the surface-based method PREDDIMER. TMH-ID was trained on the NMR and ETRA subsets and subsequently evaluated on the independent Crystal subset. For a consistent comparison, the output distance maps from the three models were converted into binary contact maps using a 3.5 Å threshold to define interfacial residues. Notably, AlphaFold2-Multimer and RoseTTAFold2 are pretrained on large-scale datasets of protein sequences and structures and thus require no additional training. Similarly, PREDDIMER is a training-free method based on surface complementarity and molecular dynamics principles, and does not involve any model fitting.

Performance evaluation using the F1 score across the 21 homodimeric TM sequences in the Crystal subset reveals that TMH-ID outperforms the three baseline models on most sequences (Figure 7). Specifically, TMH-ID achieves the highest F1 score in the majority of cases (10 out of 21 sequences), while PREDDIMER ranks second, producing more accurate predictions in six instances. AlphaFold2-Multimer surpasses the other models in only four sequences. The mean F1 score across the entire Crystal subset further confirms the superiority of TMH-ID, which exceeds the performance of RoseTTAFold2, AlphaFold2-Multimer, and PREDDIMER by 185.04%, 32.97%, and 13.44%, respectively (Table 3).

## 4. Conclusions

This study presents a comprehensive evaluation of current computational methods for predicting interface residues in α-helical transmembrane protein homodimers, a critical yet understudied problem in structural bioinformatics. We evaluated various multimer structure predictors and helical-interface-based models, highlighting their limitations in this context. The results reveal key limitations in the ability of state-of-the-art multimer models, including AlphaFold2-Multimer and RoseTTAFold2, to accurately identify interface residues in alpha-helical TM homodimers, intensifying the importance of developing specialized tools for membrane-embedded proteins. In addition, we show that the surface-based predictor PREDDIMER demonstrates superior performance compared to the other two state-of-the-art attention-based models. Building upon these findings, we introduced TMH-ID, a novel machine learning model designed specifically to predict interface residues in alpha-helical TM homodimers at high accuracy. The model integrates various sequence-based features, including large protein language model coupling scores and TM-specific motifs, in addition to structure-based features extracted from the PREDDIMER predicted structure. Notably, our proposed model, TMH-ID, outperforms the state-of-the-art machine learning model THOIPA and two other protein language models in predicting interface residues. Despite using a relatively small number of features, TMH-ID achieved the highest mean F1 score (43.9%), outperforming several advanced baselines: THOIPA (40.9%), MSA Transformer (37.7%), and ProteinBERT (34.2%). Moreover, TMH-ID outperforms other multimer structure predictors RoseTTAFold2, AlphaFold2Multimer, and PREDDIMER in interface residue prediction across the Crystal subset, achieving the highest F1 score in most cases and surpassing these baseline models by 185.04%, 32.97%, and 13.44%, respectively. We anticipate that TMH-ID’s improved accuracy would offer a valuable step forward for the membrane protein modeling community and ultimately advance our understanding of bitopic protein interactions within the membrane. While this method is developed for membrane protein homodimers, in theory it can be applied to general protein–protein interaction as well, should the same features used here be attained in these cases. It is a subject for future work to investigate how general protein–protein interaction prediction can benefit from these ideas and improvements.

## Figures and Tables

**Figure 1 ijms-26-04270-f001:**
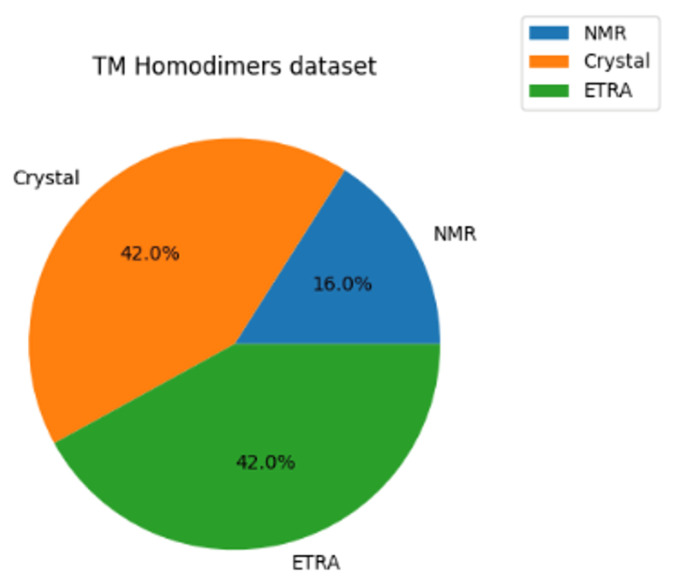
A pie chart showing the dominance of the Crystal and ETRA subsets. They represent 84% of the whole dataset.

**Figure 2 ijms-26-04270-f002:**
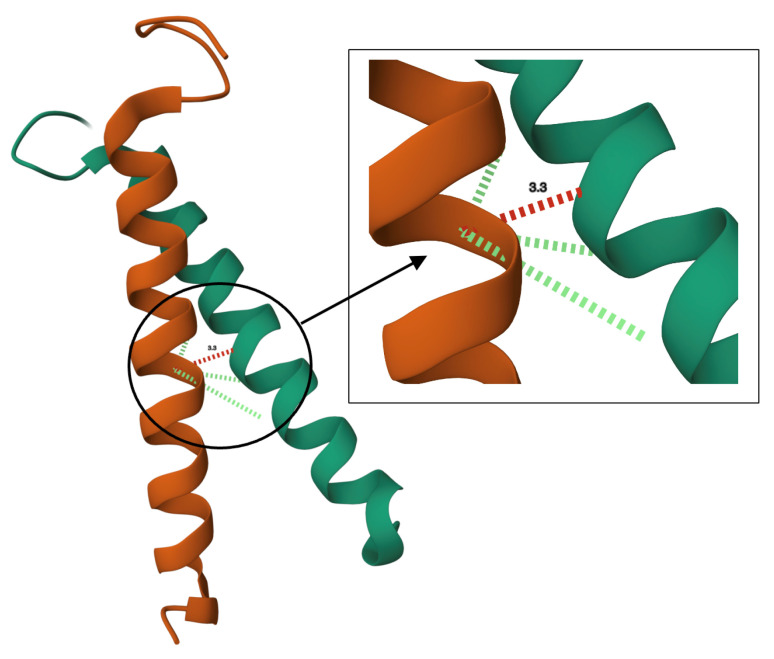
An illustration of the interface residue in a TM homodimer. The red dots show that the distance between the 2 residues is less than 3.5.

**Figure 4 ijms-26-04270-f004:**
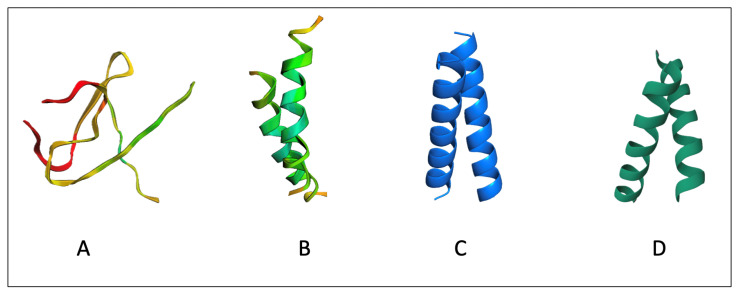
The predicted structure of PDB (5NKQA3 [40]) homodimer using (**A**) RoseTTAFold2, (**B**) AlphaFold2Multimer, and (**C**) PREDDIMER, compared to (**D**) the ground truth structure. RoseTTAFold2 failed to predict the secondary structure alpha-helices and generated random coils instead. AlphFold2Multimer on the other side successfully predicted the secondary structure but failed to identify any interface residues. PREDDIMER was able to predict the secondary structure correctly and identify some interface residues. Out of the three, PREDDIMER generated the closest structure to the ground truth.

**Figure 5 ijms-26-04270-f005:**
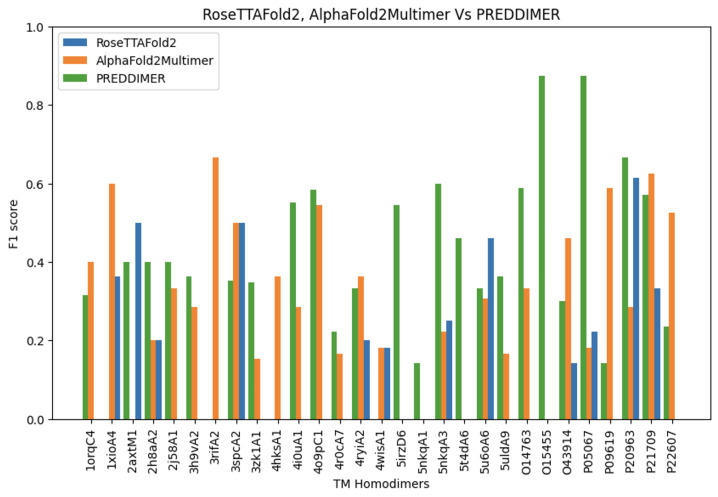
The F1 score of each individual TM homodimer separately as measured by the three models (RoseTTAFold2, AlphaFold2Multimer, and PREDDIMER). The figure shows the superiority of PREDDIMER in predicting the interface residues in TM proteins homodimers.

**Figure 6 ijms-26-04270-f006:**
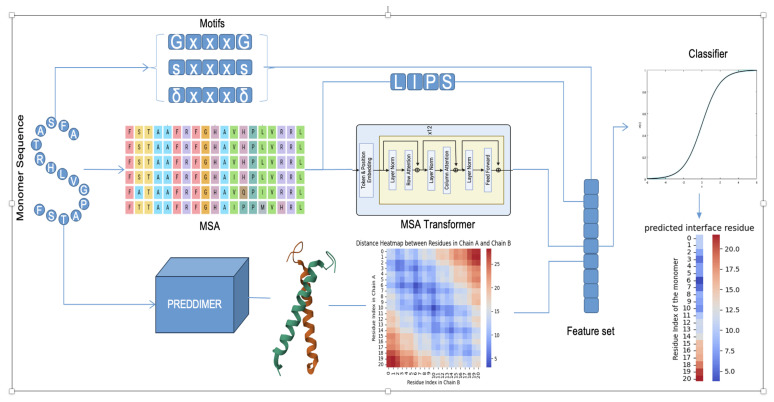
TMH-ID architecture. The model takes monomer sequence as an input. Then, it extracts motifs and features, multiple sequence alignment (MSA), and PREDDIMER dimer structure. The MSA is used to extract MSA transformer and LIPS features. The PREDDIMER structure is used to extract the distance map between residues located in different helices. These features are then fed to a logistic regression classifier to predict the interface residues.

**Figure 7 ijms-26-04270-f007:**
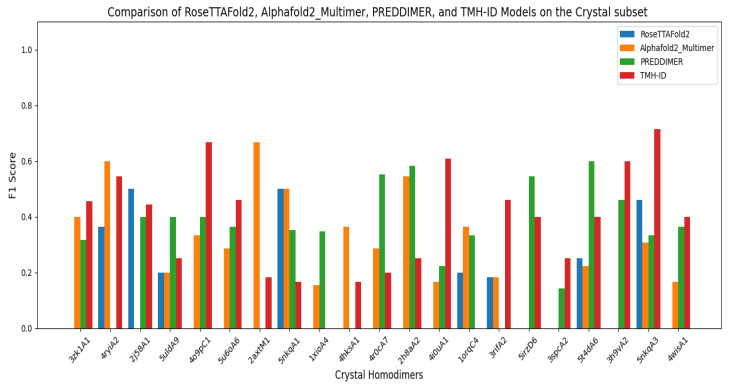
The F1 score of each individual TM homodimer in the Crystal separately, as measured by four models (RoseTTAFold2, AlphaFold2Multimer, PREDDIMER, and TMH-ID). The figure highlights the superiority of TMH-ID in predicting the interface residues in TM protein homodimers.

**Table 1 ijms-26-04270-t001:** A comparison between the three models (RoseTTAFold2, AlphaFold2Multimer, and PREDDIMER) in predicting the interface residue in TM protein homodimers using the mean precision, recall, and F1 scores.

Model	Mean Precision	Mean Recall	Mean F1 Score
RoseTTAFold2	0.111±0.1444	0.211±0.2673	0.137±0.1861
AlphaFold2Multimer	0.373±0.1661	0.270±0.2381	0.302±0.1887
PREDDIMER	0.441±0.2716	0.399±0.2421	0.378±0.2294

**Table 2 ijms-26-04270-t002:** A comparison of three machine learning based models (THOIPA, ProteinBert, and MSA-Transformer) vs. TMH-ID in predicting the interface residue in TM homodimers.

Model	Mean Precision	Mean Recall	Mean F1 Score
THOIPA	0.3411±0.1299	0.5236±0.0522	0.4085±0.0455
ProteinBert	0.3003±0.0361	0.4140±0.1227	0.3420±0.0614
MSA-Transformer	0.3168±0.0258	0.4757±0.0871	0.3774±0.0498
TMH-ID (ours)	0.3628±0.0265	0.5686±0.1121	0.4388±0.0410

**Table 3 ijms-26-04270-t003:** A comparison of three multimer structure prediction models—RoseTTAFold2, AlphaFold2-Multimer, and PREDDIMER—with TMH-ID in predicting interface residues within the Crystal TM homodimer subset.

Model	Mean Precision	Mean Recall	Mean F1 Score
RoseTTAFold2	0.11±0.1433	0.161±0.2364	0.127±0.1850
AlphaFold2Multimer	0.246±0.1492	0.331±0.2363	0.273±0.1910
PREDDIMER	0.370±0.2680	0.337±0.2309	0.320±0.1908
TMH-ID	0.362±0.1803	0.391±0.2181	0.363±0.1891

## Data Availability

The data used in this work were retrieved from [25] and are publicly available at https://osf.io/txjev/ (accessed on 16 April 2025). The proposed model code, features, and weights are available at https://github.com/Bander-Almalki/TMH-ID (accessed on 16 April 2025).

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
