# Peer review of "Transmembrane Homodimers Interface Identification: Predicting Interface Residues in Alpha-Helical Transmembrane Protein Homodimers Using Sequential and Structural Features"

_ijms, 2025, doi:10.3390/ijms26094270_

Round 1

Reviewer 1 Report

Comments and Suggestions for Authors

Almalki and Liao provided a method for extracting sequence features from the predicted structure and various domains. This study is pilot and intriguing. However, I do not think this study is thoughtful. First, this study lacks sufficient relevance because machine learning-based screening tools that only consider sequence variation and ignore physical and chemical metrics such as isoelectric points have been very widely studied. Second, the data of this study was not well disclosed, including the specific data used, the model code, and the testing situation, so the credibility and authenticity of this study has to be considered. I am regret to reject this research until the reliable information is open.

Author Response

The authors are grateful for the invaluable comments. Our responses are below.

Comment1: First, this study lacks sufficient relevance because machine learning-based screening tools that only consider sequence variation and ignore physical and chemical metrics such as isoelectric points have been very widely studied.

Response1: Although most machine learning tools do not directly and explicitly utilize physical/chemical properties, in certain way, they are incorporated implicitly via sequence variation in supervised learning. Machine learning based models have shown to be an excellent alternative to the time consuming and costly molecular dynamics models, as demonstrated by the success of AlphaFold.

This study considers not only sequence-based features to identify interface residues, but also incorporate structure-based features that can help to reduce intra-chain contacts’ noise. The features extracted from sequence not only provide information about each residue but also the context and the neighboring residues. Actually, chemical based features such as residues’ polarity are being used in the pipeline of our method.

We appreciate that the reviewer mentioned isoelectric points as widely used properties in studies of protein structure. While the isoelectric point can indeed be helpful in identifying protein-protein interaction and protein-ligand interactions by identifying the electrical charge of each unit, protein homodimers present a unique challenge. That is, the two units (chains) in the homodimers have identical sequences resulting in for the same isoelectric point (pI) score for both. This makes global pI a non-distinguishing feature for predicting inter-chain residue contacts in homodimers. Also, the isoelectric point (pI) can be useful in finding the stable structure. However, the goal of this study is not aimed at identifying the most stable structure of the dimers, but rather identifying residues that can participate in the interactions (dimerization) as they present hotspot for targeted therapies and any error in this dimerization interfaces can cause detrimental effect to the cell.

Comment2:  Second, the data of this study was not well disclosed, including the specific data used, the model code, and the testing situation, so the credibility and authenticity of this study has to be considered.

Response2: The used data is disclosed in the dataset section and in the data availability section. The code and specific data for the experiments, along with a readme file on how to reproduce our results, are now available for download at github. We hope this helps address the reviewer's concerns.

Reviewer 2 Report

Comments and Suggestions for Authors

The article titled ” TMH-ID: Predicting Interface Residues in Alpha-Helical Transmembrane Proteins Homodimers Using Sequential and Structural Features” show an interesting study comparing the developed tool with the available tools for predicting membrane positions of bitopic transmembrane proteins. This will be an useful tool to predict novel protein structures.

Comments are

  • Line 53: mention that Numerous studies have been devoted to “investigate or study” dimerization of bitopic TM proteins…

Please correct for the meaning.

  • Briefly what are the lipid data set present in the LIPid facing Surface tool.
  • Please cite the reference for ground truth structure of PDB(5NKQA3).
  • Please describe that the authors have compared TMH-ID with RoseTTAFold2, AlphaFold2Multimer and PREDDIMER in the abstract.
Comments on the Quality of English Language

GIve more explanation of how TMH-ID outperforms the compared softwares.

Author Response

The authors are grateful for the invaluable comments. Our responses are below.

Comment1:  Line 53: mention that Numerous studies have been devoted to “investigate or study” dimerization of bitopic TM proteins…

Response 1: This is fixed.

Comment2: Briefly what are the lipid data set present in the LIPid facing Surface tool.

Response2:  The tool was evaluated on a dataset of 18 TM proteins with numbers of helices ranging from 4 to 24. This info is updated in the revision.

Comment3: Please cite the reference for ground truth structure of PDB(5NKQA3).

Response3: It is cited in the revision.

Comment4: Please describe that the authors have compared TMH-ID with RoseTTAFold2, AlphaFold2Multimer and PREDDIMER in the abstract.

Response4:  We added table 3 and figure 7 to detail the comparison, and mentioned the comparison in the abstract.

Comment5: GIve more explanation of how TMH-ID outperforms the compared softwares.

Response5: Yes, we elaborated more in the revision about the rationales of our TMH-ID, which provide reasons why it outperforms other methods. In essence, our model leverages and integrates information and features that are not fully used by others, in particular, these features from predicated structures.

Reviewer 3 Report

Comments and Suggestions for Authors

The manuscript is well-written and organized. However, several comments and issues must be addressed before any final decision:

  1. The conclusion section is too short. All achievements and significant outcomes of this study should be highlighted in this section.

  2. What standard was used for the measurements in the case of the quantitative data reported in Tables 1 and 2? Additionally, error bars must be added in Figures 3 and 5.

  3. What threshold is used to classify a residue as an interface residue?

  4. Could the introduced approach be applied to other protein-protein interaction predictions? The authors should clarify this for the readers.

  5. In numerical methods, the distances between different materials (phases) are important when measuring forces and interactions. The authors should discuss this issue in the introduction section. Additionally, a discussion on different materials could be useful in providing readers with a comprehensive overview. As an example in the case of metal/metal systems the interphases distance is 1 nm as discussed in:                                  Kardani, A., Montazeri, A. & Urbassek, H.M. Computational Analysis of the Mechanical Properties of Ta/Cu Nanocomposite Dental Implants: On the Role of Incoherent Interfaces. Met. Mater. Int. 29, 2385–2397 (2023)

Author Response

The authors are grateful for the invaluable comments. Our responses are below.

Comment1: The conclusion section is too short. All achievements and significant outcomes of this study should be highlighted in this section.

Resonse1: We have lengthened the conclusion section in the revision to highlight what’s accomplished in this study.

Comment2: What standard was used for the measurements in the case of the quantitative data reported in Tables 1 and 2? Additionally, error bars must be added in Figures 3 and 5.

Response2: The results in Table 1 are the average performance for these 29 dimers in the Crystal and NMR datasets. Table 2 are average performance from 5-fold cross-validation for 50 dimers from 3 datasets: Crystal, NMR, ETRA. We have added the standard deviation to Table 1 and Table 2.

Figure 3 shows the sequence length and number of interfacial residues for these 29 dimers and Figure 5 shows f1 score for each dimer by three different methods, in order to provide more details for the comparison. Since each bar in these two figures represents just a single value -- either sequence length or a f1 value – instead of an average value, there are no error bars in these bar charts.

Comment3: What threshold is used to classify a residue as an interface residue?

Response3: 5A is the threshold for interface residue, which is adopted from the reference as cited in the manuscript: Xiao, Y.; Zeng, B.; Berner, N.; Frishman, D.; Langosch, D.; Teese, M.G. Experimental determination and data-driven prediction 437
of homotypic transmembrane domain interfaces. Computational and Structural Biotechnology Journal 2020, 18, 3230–3242.

Comment4: Could the introduced approach be applied to other protein-protein interaction predictions? The authors should clarify this for the readers.

Response4: In theory, this method can be applicable to other protein-protein interaction prediction, should the same features used from here are extracted for each interacting proteins. But this hasn’t been test, and can be a work in the future. This is clarified in the conclusion section in the revision.

Comment5: In numerical methods, the distances between different materials (phases) are important when measuring forces and interactions. The authors should discuss this issue in the introduction section. Additionally, a discussion on different materials could be useful in providing readers with a comprehensive overview. As an example in the case of metal/metal systems the interphases distance is 1 nm as discussed in:                                  Kardani, A., Montazeri, A. & Urbassek, H.M. Computational Analysis of the Mechanical Properties of Ta/Cu Nanocomposite Dental Implants: On the Role of Incoherent Interfaces. Mater. Int.29, 2385–2397 (2023)

Response5: The suggested paper is now cited in the revised manuscript.

Round 2

Reviewer 1 Report

Comments and Suggestions for Authors

Since the data and preliminary methodology of this study are heavily based on previous studies, I needed sufficient citation information to infer the correctness of the authors. In the preview version, citations in the text are shown as [? , and therefore I could not be specific in my judgment. I would recommend the authors and editors to negotiate a correction.

Author Response

Comments: Since the data and preliminary methodology of this study are heavily based on previous studies, I needed sufficient citation information to infer the correctness of the authors. In the preview version, citations in the text are shown as [? , and therefore I could not be specific in my judgment. I would recommend the authors and editors to negotiate a correction.

Response:  The many question marks in the citation were due to an incorrect rendering of the PDF file from the submission system. The problem was fixed in an updated version we emailed the editorial office afterwards. Our apology for the inconvenience.

Round 3

Reviewer 1 Report

Comments and Suggestions for Authors

The author provided a efficient, accurate and effective numeric predicted model. The authors has revised the manuscript according to what I concerned. I am pleased to taken the whole review process and enlighten to recommend its acceptance in IJMS.